# The Importance of Gender-Sensitive Health Care in the Context of Pain, Emergency and Vaccination: A Narrative Review

**DOI:** 10.3390/ijerph21010013

**Published:** 2023-12-21

**Authors:** Joachim Graf, Elisabeth Simoes, Angela Kranz, Konstanze Weinert, Harald Abele

**Affiliations:** 1Institute for Health Sciences, University Hospital Tuebingen, Midwifery Science, Hoppe-Seyler-Str. 9, 72076 Tuebingen, Germany; angela.kranz@med.uni-tuebingen.de (A.K.); konstanze.weinert@med.uni-tuebingen.de (K.W.); harald.abele@med.uni-tuebingen.de (H.A.); 2Department for Women’s Health, University Hospital Tuebingen, Calwerstr. 7, 72076 Tuebingen, Germany

**Keywords:** gender-sensitive health care, sex and gender differences, narrative review, pain, emergency, vaccination

## Abstract

So far, health care has been insufficiently organized in a gender-sensitive way, which makes the promotion of care that meets the needs of women and men equally emerge as a relevant public health problem. The aim of this narrative review was to outline the need for more gender-sensitive medical care in the context of pain, emergency care and vaccinations. In this narrative review, a selective search was performed in Pubmed, and the databases of the World Health Organization (WHO), the European Institute for Gender Equality and the German Federal Ministry of Health were searched. Study data indicate that there are differences between men and women with regard to the ability to bear pain. On the other hand, socially constructed role expectations in pain and the communication of these are also relevant. Studies indicate that women receive adequate pain medication less often than men with a comparable pain score. Furthermore, study results indicate that the female gender is associated with an increased risk of inadequate emergency care. In terms of vaccine provision, women are less likely than men to utilize or gain access to vaccination services, and there are gender-sensitive differences in vaccine efficacy and safety. Sensitization in teaching, research and care is needed to mitigate gender-specific health inequalities.

## 1. Introduction

### 1.1. Gender and Public Health: Background

Disease occurrence of men and women differs with regard to epidemiology, manifestations and causes in many diseases. Accordingly, the implementation of gender-sensitive health care is necessary which adequately considers gender-related differences in prevention, diagnostics, therapy and care. In this context, the effects of both biological and social gender must be taken into account [1,2]. However, in many member states of the World Health Organization (WHO), health care has so far been insufficiently gender-sensitive [3,4]. This makes the promotion of gender-sensitive care a relevant public health problem, especially since gender equality in medical care corresponds to the core demands of the United Nations (UN) and the WHO (e.g., in the context of the Sustainable Development Goals) [5]. From a public health perspective, gender-sensitive health care also refers, in particular, to access to health services, gender-specific differences in risk-related behavior (e.g., related to smoking [6]), the gender-sensitive organization of prevention and health promotion, and the influence of gender on health determinants [7], while gender medicine seeks to orient the care of individual patients also to their gender-specific needs in the context of personalized medicine [8]. Both the individual medicine and public health perspectives are related in the sense that gender-sensitive personalized medicine requires corresponding anchoring in the public health framework. In this regard, the consideration of sex and gender in public health research is essential to optimize methodological approaches, close the gender gap in public health knowledge and advance gender equality. However, the study findings here suggest that gender aspects have been insufficiently addressed in public health research proposals and that public health inadequately addresses gender disparities [9]. The COVID-19 pandemic exacerbated gender inequalities [10,11], in part because pandemic interventions were not sufficiently gender-sensitive [11,12].

### 1.2. Aims

Against this background, this narrative review outlines the need for more gender-sensitive medical care using the examples of pain, emergency care and vaccinations. The aim is to highlight the relevance of gender-sensitive medicine from a women’s health perspective and to identify gender medicine issues as relevant areas of teaching and research for women’s health and midwifery science [2,8]. The examples were chosen because they are highly relevant to women’s health and obstetrics. Studies suggest a misuse and underuse of pain management in women in general [13] and in the context of childbirth in particular [14,15]. With regard to emergency care, gender effects are also likely to be influential factors in the overall care algorithm [16]. Emergency situations can also occur during pregnancy, although the extent to which gender stereotypes influence prehospital care or its duration here has not yet been investigated. Srajer et al. (2023) found that emergency physician gender is associated with early pregnancy loss management [17]. Analogous findings apply to immunization as a significant public health strategy; using a regression model, Fuertes et al. (2022) showed that gender inequality at the national level is predictive of childhood immunization coverage, indicating that removing gender barriers is essential to achieve universal immunization coverage [18]. The case studies were identified as particularly relevant during a discussion with midwifery students in the course “Gender Medicine and Women’s Health” (publication in preparation) and are interlinked in terms of content: non-gender-sensitive pain care is likely to be particularly evident in emergency care, and gender-specific differences in the efficacy and side-effect profile of analgesics are also evident in the case of vaccines.

This leads to the following research questions: How do biological and social gender affect pain, emergency and vaccine care? How does biological sex influence pain perception, as well as efficacy, responsiveness and side-effect profiles for pain medications and vaccines? In which sense does social gender or existing gender stereotypes influence the likelihood of receiving appropriate emergency care? To what extent do gender issues influence adequate care and utilization of pain medications and vaccines? We will examine the extent to which pain, emergency and vaccine care are gender sensitive.

## 2. Methods

In this narrative review, a selective search was performed in Pubmed, and the databases of the WHO, the European Institute for Gender Equality, and the German Federal Ministry of Health were searched. The search terms “pain”, “analgesics”, “emergency care”, “vaccination”, “vaccine safety”, “vaccine efficacy”, “side effects”, “gender effects”, “sex effects”, gender equality” and “gender-sensitive” were combined via Boolean operators AND and OR. English- and German-language studies that were not older than 15 years as well as textbooks, dissertations and legal texts were considered (inclusion criteria). In addition, current textbooks on the topic of gender-specific medicine and women’s health were screened for content, and the literature cited here was reviewed (principle of snowball search). With regard to the legal location of gender-sensitive health care in the German health care system, the German Social Code Book V was also screened. Excluded were texts written in languages other than German or English and gray literature publications. Overall, 250 documents were identified and screened (either abstract or full text), of which more than half were excluded due to being unsuitable. A further restriction was not possible but also not desired, as the topic should be covered as broadly as possible. It became apparent that there is a lack of reviews on the topic and that some of the individual studies only contained fragmentary information.

## 3. Gender Medicine and Gender-Sensitive Care

Gender medicine is a discipline investigating gender-specific aspects of health and illness (e.g., with regard to gender-specific differences in symptoms, self- and external assessment and communication of patients, as well as diagnostics and care in outpatient and inpatient settings). It refers to both biological (sex) and social (gender) aspects and is interested not only in biologically based differences but also in socially constructed causes of illness and health, related to, for example, gender stereotypes, which, among other things, influence risk and health behavior. Gender as a social structural category is thereby not only related to individuals but also determines social value systems as a social structural category [19]. The WHO emphasizes gender as one of the central social determinants of health [20], since gender roles and norms, gender-based discrimination and violence, and structural gender (in)equality have influence on care [21]. Glezerman defines the central aim of gender medicine *“to acknowledge the physiological and pathophysiological differences between men and women in the treatment of their bodies”,* whereby *“the existence of these differences requires that the medical profession awaken to the fact that men and women have different needs when it comes to their health”* [22]. As Oertelt-Prigione emphasized, *“sex- and gender-sensitive medicine focuses on the role that biological differences (sex) and socio-cultural power structures (gender) play in healthcare”* [23]. It is assumed that biological and social sex are not diametrically opposed, but that there is a seamless continuum between the two domains, in the sequence genetic sex (according to the possession of sex chromosomes), gonadal sex (according to the formation of the gonads), genital sex (according to the physical external sex characteristics), psychological sex (the gender identity as self-identification) and social sex (gender and externally originating social assignment of gender roles). All dimensions are important in order to capture differences in disease development, symptomatology and expression in a gender-sensitive way [19,21,22,23]. To fully consider the impact of gender on health and health care, the intersectionality perspective is of relevance [24].

The underlying understanding of a continuum consisting of sex and gender effects on health [22] can be seen in the case of cardiovascular diseases. Although myocardial infarction, for instance, occurs with similar frequency in women and men [25], diseases of the cardiac vessels (e.g., CHD) function as “male” phenomena, with the consequence that they are considered underdiagnosed in women (gender effect), and because women and men differ in terms of causes and symptoms due to biological differences [26]. The connection between gender and sex effects leads to the fact that women with acute coronary symptoms are less often treated according to guidelines and have a higher mortality rate than men of the same age [27]. Gender medicine emphasizes that, for example, differences in disease etiology and symptomatology must be considered separately for all genders in order to provide adequate care, thus focusing on areas of biological and gender inequality [22,28]. In myocardial infarction, this is reflected, among other things, in the fact that females present on average with more atypical symptoms such as nausea, vomiting and shortness of breath [29].

In order to develop gender-sensitive health care, various communication and promotion strategies must be implemented [30]. A central level is represented by health policy, which expands the legal requirements of the health care system to include appropriate aspects [30,31]. Germany has explicitly complied with this requirement within the framework of the Prevention Act of 2016, which states that health risks are unequally distributed, are partly caused by the social environment and are reproduced by social inequality, and which has also become established in the German health care system [32]. § 20 of the German Social Code Book V explicitly recognizes the importance of gender for health inequalities. It states that *“the services* [provided by the statutory health insurance funds in the context of prevention and health promotion] *should contribute in particular to reducing socially conditioned as well as gender-related inequalities in health opportunities”* [33]. This focuses on core aspects of gender medicine, which advocates taking gender into account as a central factor in explaining differences in the occurrence and experience of illness and health [19].

## 4. Case Study: Pain

### 4.1. Gender-Specific Pain Perception and Communication

Gender effects can be found in pain care [34]. Sex and gender effects are also evident in pain. It is assumed that there are differences in the perception of pain and the ability to endure pain, although it must also be taken into account that the survey of pain has not yet been carried out in a sufficiently gender-sensitive manner. So far, gender-related differences in the pain threshold are insufficiently researched. On the other hand, socially constructed role expectations in the case of pain as well as the communication of the same are relevant, both in self-assessment and in assessment by others. Pain experience is highly subjective and consists of sensory, cognitive and emotional components [34]. Consequently, several dimensions must be distinguished here through which gender can influence the experience of pain. Recent studies have revealed significant gender differences in the physiological mechanisms underlying pain, including the gender-specific involvement of different genes and proteins, and different interactions between hormones and the immune system that influence the transmission of pain signals [34,35,36,37]. Neuroimaging in humans has revealed sex-specific differences in neural circuits associated with pain, including sex-specific brain changes in chronic pain conditions. Clinical pain research suggests that gender may influence how a person contextualizes and manages pain. Gender may also influence vulnerability to the development of chronic pain [35,36,37]. Studies suggest that men and women respond differently to pain, with women generally having a higher risk for clinical pain. Differences in responsiveness to pharmacological and non-pharmacological pain treatments have also been observed; emerging evidence suggests that genotype and endogenous opioid function play a causal role in these differences, and there is ample evidence in the literature that sex hormones influence pain sensitivity [37,38,39]. Psychosocial processes such as pain coping and early exposure to stress may also explain differences between the sexes, in addition to stereotypical gender roles that may contribute to differences in pain expression [38]. Furthermore, Goffaux et al. (2011) hypothesized that sex differences in anxiety would explain sex differences in experienced pain and physiological responses to pain (at both spinal and cortical levels) [39]. Pain in women is more likely to be perceived and dismissed as “contrived or exaggerated”. Gender effects influence how reactions to pain manifest. This is also valid with regard to the readiness to act that follows an expression of pain and also the pain indication itself—according to the results of a meta-analysis [40].

### 4.2. Pain Management Depending on Gender

Several studies have recently identified gender effects in the provision of pain medication, with significant differences in patients [37,38,39]: for example, a prospective cohort study that investigated the type and duration of pain medication. A total of 981 patients were consecutively included who presented to the emergency department with acute abdominal pain. With comparable pain scores, fewer women than men received adequate pain medication (60% vs. 67%). Of relevance was that the gender of the attending physicians did not influence the likelihood of receiving adequate pain medication, as women were significantly less likely to be prescribed an analgesic than male patients by both male and female physicians [34]. Another study found that when pain medication was administered by the ambulance service before transfer to hospital, women were significantly less likely to receive morphine than men [41]. More fundamentally, study findings suggest that the burden of chronic pain is greater in women than in men: an estimated 54.9% of women versus 48.5% of men in the US suffered from chronic pain in 2010. However, although women report higher pain scores, they receive less intensive and effective pain management than men, but more frequent antidepressants and referrals to mental health care [42] than men in comparable situations. Overall, there are gender biases in the timing, amount and type of pain treatment administered, which means that women are more likely to receive inadequate care [43].

In addition to the aforementioned gender aspects, sex effects must also be taken into account in pain care. For example, 37% of a total of 67 agents approved by the FDA between 2000 and 2002 showed gender-specific differences in terms of pharmacokinetics, efficacy or side effects [44]. For example, both biological gender differences and socio-cultural gender norms are not sufficiently taken into account in pharmaceutical safety for women. There is also evidence that women respond more slowly to intravenous morphine administration and that the effects last less time than in male patients [45].

Examples from obstetrics show that the factor of migration interferes with pain relief measurements for women in labor; women with a migration background are less ready to use pain medication in the context of childbirth or are offered it less often than women without migration status due to different pain perception, coping and communication [46], although a higher pain burden due to trauma, violence, stress and refugee experience has been proven for migrant women [47]. Figure 1 shows a simplified representation of sex and gender effects on pain care.

## 5. Case Study: Emergency Care

### 5.1. Gender Inequality in Emergency Care

Study results suggest that the female gender is associated with an increased risk of inadequate emergency care. The aim of a Swedish study with *n* = 383 patients was to investigate whether gender differences exist in the prehospital care of severely injured trauma patients. Male patients with a comparable degree of injury in relation to females were 2.75 (95% CI: 1.2–6.2) more likely than female patients to receive the highest priority level of prehospital care when the mechanism of injury and vital signs at the scene were taken into account. Men were significantly more likely to receive priority one care from paramedics and were significantly more likely to be transported directly to a trauma unit, according to the key findings of this study [48]. In a cohort study of 26,861 trauma patients which aimed to investigate the influence of gender on access to care in a trauma center, similar gender differences were found. Women with a comparable degree of injury were significantly less likely than men (OR = 0.87; 95% CI: 0.79–0.96) to be treated at a trauma center (for the same injuries) and significantly less likely to be transported from the scene to a trauma center, regardless of whether the initial care was provided by paramedics (OR = 0.88; 95% CI: 0.81–0.97) or physicians (OR = 0.85; 95% CI: 0.73–0.99) [49]. Both studies did not consider the gender of the practitioners. However, it is known from other aspects of care that the influence of the gender of treating physicians on the outcomes is significantly lower than the gender of the patients [34].

### 5.2. Emergency Care and Gender Stereotypes

As a consequence, it seems to be probable that the observed gender differences in access to care in trauma centers are multifactorial. Conscious or unconscious gender bias on the part of treating physicians was also identified as a possible cause for the observed gender differences. Borkhoff et al. investigated whether the decision to refer patients for or to perform total knee arthroplasty was influenced by their gender [50]. After analyzing data from standardized patients with identical clinical scenarios who differed only in terms of gender, both general practitioners and orthopedic surgeons were more likely to recommend total knee arthroplasty to men than to women. However, it remained unclear whether the decision was based on conscious discrimination, on an unconscious assumption of the potential benefit or as result of the communication process (for example, because male patients emphasized more strongly that they were dependent on the full functionality of the knee due to occupational activity or sport [50]). Gender stereotypes, i.e., the attribution of characteristics to a gender category, have an influence on the non-individualized treatment of women and men. These stereotypes are learned in the course of socialization and are often unconsciously reproduced daily in social discourses [51]. Male gender stereotypes include the assumption that men are often in a health-wise worse condition than they admit and therefore are supposed to require higher priority trauma care in the event of an accident [52]. In fact, it is known from studies that the male gender is associated with a higher risk of accidents and consequently also with a higher risk of trauma [52]. However, the care in the case of an accident is only one part of the care repertoire of emergency medicine. A retrospective study from the University Hospital of Düsseldorf showed that of 43,821 patient contacts in the central emergency department of the university hospital, the proportion of female patients was 48%; so, women and men do not differ in terms of the probability of needing emergency medical care [53]. One of the stereotypes concerning the female gender is the assumption that women exaggerate their symptoms and are therefore less ill than they communicate. As a result, women’s complaints (with the same symptoms) are more often diagnosed as psycho-vegetative disorders and treated by prescribing psychoactive substances, and women with the same symptoms more often receive a psychological diagnosis, while men receive a somatic diagnosis [54]. For emergency care, this means that women with, e.g., suspected myocardial infarction have a significantly lower priority for the emergency service than men, which is associated with a significantly longer total ischemia time and correspondingly less favorable outcomes (related, e.g., to mortality), as was found in a study from Norway [55]. Gender stereotypes lead to gender-specific categorization, which as an adaptive strategy aims to release cognitive resources by simplifying complex situations to enable rapid decision-making. In the context of outpatient care, in situations of high stress and time pressure for ambulance personnel, patients need to be quickly categorized into treatment pathways that determine what type of care they will receive. Gender stereotypes and the resulting bias can lead to women being placed in less urgent or even non-trauma-specific care pathways [49,56]—i.e., with a mis- and under-supply of care and less advantageous treatment options.

### 5.3. Consequences of Non-Gender-Sensitive Emergency Care

A less correct prioritization in emergency care often results in poorer outcomes for women. For example, the female gender is associated with higher mortality among trauma patients, even though women are less likely to be involved in serious accidents than men. Female trauma patients have more severe head injuries and more spinal and pelvic injuries and are more likely to be trapped in car accidents than male patients, partly because the ergonomics of car seats are generally designed to fit the physiology of the male body rather than female proportions [57,58]. In the context of emergency care, women are then less often assigned the highest priority and are less often given guideline-compliant trauma care; they even receive less often basic measures such as infusions than male trauma patients [49,58]. In cases of suspected myocardial infarction, it was found that the arrival of an ambulance or first medical contact after making the emergency call takes significantly longer if the person making the emergency call is female [59].

Against this background, it is of special interest to consider acute care areas in the clinical and outpatient context that are ostensibly open almost exclusively to women, such as the delivery room or birth centers. Here, too, gender effects must be discussed and taken into account. Gender effects have an influence on the entire care algorithm, which applies similarly to pregnant women during the phase before reaching the delivery room or the emergency outpatient department. Emergency situations can also occur during pregnancy; however, it has not yet been investigated to what extent gender stereotypes influence prehospital care or its duration in this context. Figure 2 shows a simplified representation of sex and gender effects on emergency care.

## 6. Case Study: Vaccination

### 6.1. Gender-Related Differences in Utilization

Vaccination is a preventive intervention with a broad reach and significant population health benefits, as it can contribute to the control or eradication of infectious diseases (e.g., measles vaccination as a primary prevention intervention) or to the reduction in the incidence of risk factors (e.g., HPV vaccination as a primordial prevention intervention to reduce the disease burden of HPV-induced cervical cancer) [60,61]. Preventive measures are attested to be a high priority in the health care system, which is why a high utilization rate should be aimed for. Women are more willing than men to participate in preventive services, regardless of age and social status [62,63]. They are characterized by a more favorable health behavior (e.g., with regard to the use of screening and preventive measures, health courses or addictive behavior) [64]. The category of gender therefore influences health in many ways (due to different gender-related role expectations also with regard to behavior in the context of health and illness) and in the area of prevention [51].

However, the higher willingness of women to use preventive care is not mirrored in the utilization of vaccination offering.: In Germany, women have a lower vaccination rate for tetanus, diphtheria, influenza or measles [65] as well as for COVID-19 [60,66], compared to the male population, over their entire life course. For rubella, the vaccination coverage rate for women is higher than for men, but the vaccination rate for women aged 30–39 years in 2013 was only 54.5%, which increases the likelihood of vaccine-preventable rubella embryofetopathy in the case of pregnancy in this group [65]. Greater vaccination skepticism is already evident among female adolescents compared to males [67]. However, the causes for minor vaccination rates in girls and women are complex and must also be explained sociologically. Studies indicate a systematic disadvantage for girls and women when it comes to vaccination due to the gender-specific social inequality reproduced in the social subsystems. This means that the gender of the child influences the parents’ decision for or against vaccination. Gender inequalities in vaccination coverage are not only evident in less developed economies, such as India, where girls are significantly less likely to be vaccinated than boys [68]. Regarding the willingness to vaccinate their own children against COVID-19, an American study showed that fathers were significantly more willing to have their male children vaccinated (odds ratio (OR) = 1.62) than their female children [69]. Study results from Sweden reported that parents who decide against HPV immunization of their daughters justify this, among other things, by their aim to prevent the girls from having sexual contact before marriage [70]. Although it is known that a high vaccination rate against HPV in girls at the beginning of puberty more than halves the incidence of cervical carcinoma, gender-specific role expectations (also related to sexual self-determination) thus influence health-related behavior to the disadvantage of girls and women [71]. For some years, HPV vaccination has also been recommended for boys as a preventive treatment against cancers such as oropharyngeal, penile and anal cancer [72]. Here, there is also an extremely low utilization rate; however, studies have shown other possible reasons than those for girls. There are indications of a lack of knowledge among many parents who often assume that the vaccination has no intrinsic benefit for boys but only eliminates them as “HPV transmitters” for the sole benefit of girls [73].

### 6.2. Sex Differences in Efficacy

While vaccination utilization is influenced by the category of gender, there is also an impact of biological aspects in vaccine delivery—related to efficacy and potential side effects. Study results suggest that women develop higher antibody responses to many vaccines compared to men, and that the variables of age and sex interact in predicting vaccine response [74,75]. For example, higher levels of estrogen may contribute to an enhanced vaccine response in women, while testosterone correlates with an attenuated vaccine response, e.g., in influenza vaccination [76]. Genetic factors and their interactions with sex hormones are also associated with this sexual dimorphism in immunological response: there are about ten times more genes on the X chromosome than on the Y chromosome, including a large proportion of genes coding for immune-related proteins [77]. Females therefore have higher expression of immune-related genes and proteins that can interact with sex hormones to enhance an immune response [75]. In principle, gender-specific differences in antibody response (and thus related to efficacy) depend on the particular vaccine. Women of older age have higher antibody responses to influenza vaccines than men of the same age. Men of older age, in contrast, show a higher clinical efficacy with the pneumococcal vaccine, as well as with the diphtheria, tetanus and pertussis (combination) vaccines [78]. This contrasts to reports in younger adults, where women have higher antibody titers than men in response to live, subunit and inactivated attenuated vaccines—including pneumococcal, influenza, yellow fever, rubella, measles, mumps, hepatitis A and B, herpes simplex 2, rabies, smallpox and dengue vaccines [78,79,80]. COVID-19 vaccines were shown to be more effective in males in most pivotal (phase III) trials. However, age stratification was not used in most of these studies [75].

### 6.3. Adverse Events and Gender

In the context of sex-specific differences in antibody response and vaccine efficacy, there are also sex- and gender-related differences in the frequency of reported adverse events. On the whole, these are considered to be very rare events. However, it is known that women show adverse reactions after vaccination more frequently than men [71,75]. Between 1990 and 2016, women aged 19–49 years accounted for 83% of anaphylactic reactions following vaccination in the USA [81]. Data from the Centers for Disease Control and Prevention (CDC) showed that among people in the 20–59 age group who received the 2009 H1N1 vaccine, postvaccination hypersensitivity reactions were at least four times more likely to occur in women than in men, although more men received vaccination [75,82]. Corresponding sex-related effects are also characteristic for the COVID-19 vaccines. Here, epidemiological studies show that very rare severe adverse events (such as sinus vein thrombosis), which can only inadequately be measured in registration studies due to their rarity, affect women significantly more often. For the first 13.7 million COVID-19 vaccine doses administered in the US, a CDC report found that 61.2% of all 13,794,904 vaccine dose recipients were female, but 79.1% of the 6994 adverse events reported affected women [75,83]. Another study focusing on the efficacy and safety of mRNA vaccines showed that all 19 people who had adverse reactions to the Moderna vaccine were female, as were 44 of the 47 people who reported anaphylactic reactions to the Pfizer BioNTech vaccine [75,84]. Analogous reports were stated in Germany by the Paul Ehrlich Institute: for all COVID-19 vaccines licensed in Germany, suspected adverse reactions/vaccine complications were reported significantly more often in women [85].

### 6.4. Lack of Sex- and Gender-Sensitive Vaccine Research

The presented sex-specific differences in efficacy and safety would have to be considered in vaccine research and licensing studies. This also applies in particular to vaccines administered during pregnancy. Here, there is a need for improvement. Registration studies for vaccines (e.g., COVID-19) are often not sufficiently gender-sensitive when it comes to documenting side effects, as Jensen et al. point out [75]. Most pivotal vaccine trials focus on healthy subjects aged 18–65 years, excluding the elderly, pregnant women and postmenopausal women [86]. Women of childbearing age are also often excluded from clinical trials for drug and vaccine approvals for justified reasons, following the incidence of neonatal malformations in the 1970s. This was originally intended to protect women and newborns, but their exclusion from early phases of drug development led to knowledge deficits, particularly with regard to the safety profiles of drugs, with measurable consequences for women’s health [75]. Similarly, pivotal studies for COVID-19 vaccines did not consistently stratify safety data by sex [87]. In some cases, almost twice as many males as females were included in the phase III trials [88], making sex-specific documentation difficult, especially for rare adverse events. Figure 3 shows a simplified representation of sex and gender effects on vaccine care.

## 7. Discussion

### 7.1. Discussion of Overall Results

Gender, both as an individual and as a social category, has a significant influence on health outcomes, and as a core factor of the health determinants, influences all levels of this model [21,89]. Due to an existing gender-specific social inequality, the category of gender acts here as an effect amplifier—often to the unfavorable side of girls and women. We did not find any studies that postulate a gender-specific disadvantage for boys or men in the area of emergency or pain care—which does not mean that this could not theoretically exist (publication bias), as the category of gender permeates the health-care system as a multi-layered phenomenon and has the effect of reproducing gender stereotypes. For example, male gender stereotypes reproduce the idea that mental illnesses primarily affect women—with the result that depression (the clinical presentation of which is also different in men than in women) is diagnosed significantly less frequently in men [90] and, for example, postpartum depression in fathers is not sufficiently identified and treated, despite a presumably high incidence [91]. However, the focus of this narrative review was not on identifying inadequate gender-specific care for mental illnesses. In order to cushion the resulting gender-specific health inequity, awareness needs to be raised in education, research and care. The cases presented here point to existing disparities.

Personalized medicine in pain care means that gender-specific aspects must also be considered in order to be able to provide an individually adapted, efficient therapeutic regime for existing (chronic) pain [13], which also includes the realization of gender-sensitive differences in pain perception during the rehabilitation phase [92]. Studies indicate that emergency care is insufficiently organized in a gender-sensitive manner, i.e., the gender of the patient influences the probability of receiving guideline-compliant trauma care and/or the attributions of a priority level necessary for the admission to an emergency department. Furthermore, a lack of gender-specific emergency care is also a result of the fact that the emergency medical services continue to be rather male-dominated, not only in terms of the gender distribution within the professional fields involved (emergency physicians, emergency paramedics, and paramedics) but especially in terms of the standards taught in training. For example, when describing myocardial infarction, textbooks mainly refer to symptoms that appear in men [29,93], even though it has been known for a long time that men and women differ in the leading symptoms [26]. Gender stereotypes, often also unconsciously, underlie the actions of health professionals, which contribute to the fact that women are less likely to receive adequate care in emergency care. For all professional groups involved in emergency care, this results in the need to be sensitized to gender-sensitive care. This calls for reflection on how gender inequality is reproduced in everyday care [94,95]. Secondly, it needs to be communicated that women and men differ in the way that they communicate, both at the patient and at the professional level. The gender of patients influences how complaints are communicated verbally and non-verbally and thus which treatment pathways are taken [96]. At the same time, the gender of physicians influences the type of information that patients communicate in their medical history [97]. The communication skills of medical students of female and male gender already differ during their undergraduate studies [98], and similar reciprocal effects can be assumed for nursing.

Therefore, health professionals should already be sensitized to aspects of gender medicine during their studies and training periods. Against the background of a need for gender-sensitive emergency care, this applies, in particular, to emergency paramedics, emergency doctors and nurses but also to midwives. According to current UN and WHO recommendations, midwives, as experts in the care of women of childbearing age, should represent and strengthen women’s health in all areas (corresponding also to the Health in All Policies concept)—a task which explicitly requires sensitization to gender-sensitive aspects of care [99].

This also applies to preventive interventions such as vaccinations. Scientific data from health research impressively illustrate that a high utilization of vaccination offers is to be aimed for within the field of prevention. However, sex-specific differences in effectiveness and safety are evident. It must be made clear that registration studies have so far been insufficiently gender sensitive, although this is recommended in current guidelines for clinical studies. Women should be included in drug studies to an appropriate extent, and the analyses of safety and efficacy data of approved drugs should be stratified by gender [75]. In addition, a gender bias is evident in that women are disadvantaged (due to structural social inequality) in the supply of vaccines.

Gender medicine refers to the need to adequately incorporate the physiological and pathophysiological differences between men and women in care, thus emphasizing the differences between the sexes, which are also reflected in the needs within the health system [22,23]. Gender refers to the influence on the results from the interaction between the sexes or the individual–structural interaction. Men and women differ in the way that they communicate, which includes pain communication [40]. Gender is assigned complex role expectations and functions in the context of social action, which influence the social role of gender and also have an impact on the health system [51], e.g., on the probability of receiving adequate pain medication [43] or vaccination [67,68,69].

### 7.2. Answering Research Questions

The central research question was how biological and social gender affect pain, emergency and vaccine care from the perspective of women’s health. It could be shown that biological sex has an influence on pain perception as well as on the efficacy, response and side-effect profiles of pain medication and vaccines. Thus, both pain perception and pain management are influenced by sex hormones, among other factors [37,38,39]. Women respond more slowly to intravenous morphine administration and the effects last for less time than in male patients [45]. Study results suggest that women develop a stronger antibody response to many vaccines than men [74,75]. This results partly from estrogen production, as higher estrogen levels contribute to an enhanced vaccine response, and from genetic factors, as the X chromosome contains a large proportion of the genes responsible for coding immune-related proteins [75,76,77]. Sex hormones and higher expression of immune-related genes also increase the risk of adverse events after vaccination in women [75,81,82,83,84,85].

Regarding the influence of social gender and existing gender stereotypes on the likelihood of receiving appropriate emergency care, it can be concluded that male patients with a comparable level of injury are more likely to receive the highest priority level of prehospital care than women and are significantly more likely to be transported from the scene to a trauma center [34,48,49]. Women with, for example, suspected myocardial infarction show significantly longer total ischemic time and correspondingly less favorable outcomes (e.g., in terms of mortality) because of low prioritization within prehospital care [55].

With regard to the influence of gender on the provision and use of pain medication and vaccines, it was noted that women are more likely to be underserved in the provision of adequate pain medication in the clinical setting and are less likely to receive pain medication in the required dosage for abdominal pain, for example (given the same pain score) [37,38,39,40,41,42,43]. In vaccine coverage, there is evidence (e.g., for COVID-19 and HPV vaccinations) of a socially determined disadvantage for girls based on their sex [69,70]. For other vaccinations, there is evidence of lower uptake among women [65,67].

### 7.3. Conclusions

The main conclusion is that neither pain, emergency nor vaccination care has been structured sufficiently to be gender-sensitive to date. This means that aspects of misuse, underuse and overuse are evident in all three dimensions, because the influence of the category of gender is not considered and gender stereotypes are reproduced as a result. If the focus is placed on the high epidemiological significance of pain or the fact that vaccinations address the entire population as a preventive measure, it becomes clear that the health-care system plays a central role in maintaining gender-specific social inequality, which also entails high national economic costs.

### 7.4. Future Developments

It can be noted that neither pain nor emergency nor vaccine care has been sufficiently structured to be gender-sensitive to date, although gender-sensitive health care is one of the WHO’s central demands [20]. Multiple public health strategies to implement gender-sensitive care include not only anchoring it in national health systems [32,33] but also raising awareness early in medical education [30,31]. Wortmann et al. stated that gender medicine teaching increases medical students’ gender awareness [100]. In Germany, gender medicine will become a mandatory component in the medical curriculum from 2025, and faculties are accordingly called upon to develop appropriate teaching models. The case studies centered on pain, emergency and vaccination should be given central consideration because of their importance for women’s health. It would make sense to develop interprofessional teaching formats to sensitize future physicians and midwives together regarding the importance of gender-sensitive care. The aim of women’s health research must be to cushion the aforementioned gender effects in health care. In terms of technical developments in the health sector, such as artificial intelligence, it is becoming increasingly relevant to solve these challenges at an early stage. If these differences are included in the development of algorithms, this reinforces gender-specific discriminatory results [101].

## Figures and Tables

**Figure 1 ijerph-21-00013-f001:**
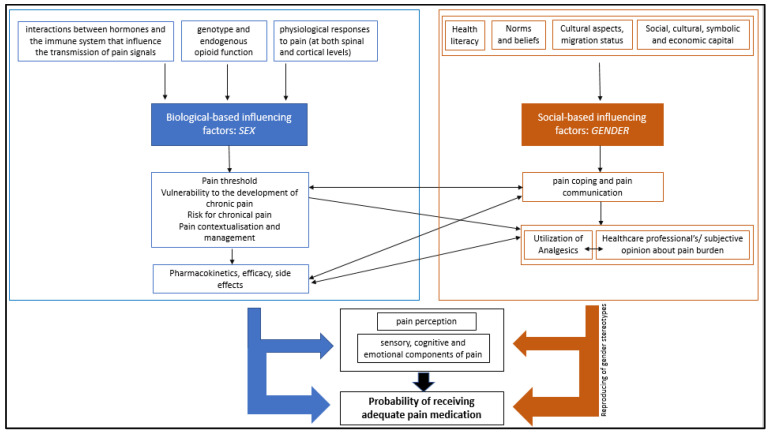
Simplified representation of sex and gender effects on pain care.

**Figure 2 ijerph-21-00013-f002:**
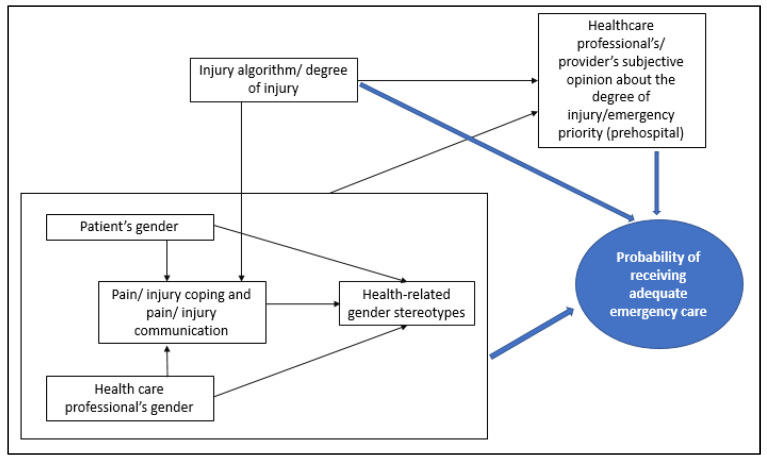
Simplified representation of sex and gender effects on emergency care.

**Figure 3 ijerph-21-00013-f003:**
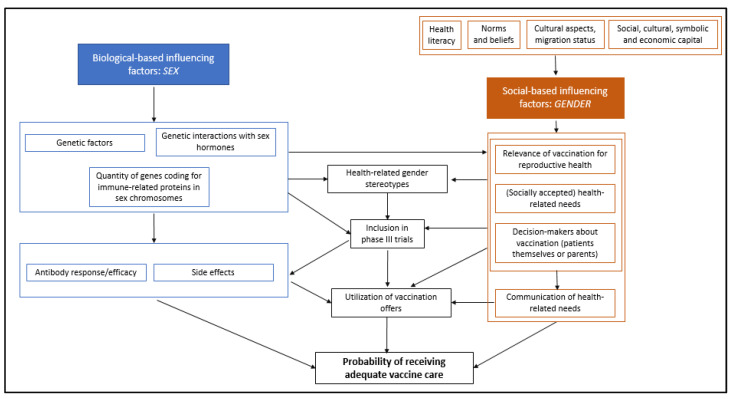
Simplified representation of sex and gender effects on vaccine care.

## Data Availability

No new data were created or analyzed in this study. Data sharing is not applicable to this article.

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
