# Peer review of "The Importance of Gender-Sensitive Health Care in the Context of Pain, Emergency and Vaccination: A Narrative Review"

_ijerph, 2023, doi:10.3390/ijerph21010013_

Round 1

Reviewer 1 Report

Comments and Suggestions for Authors

This paper presents an important contribution to the literature, and can serve as an important reference for those in need of a broad overview of the role of gender in medicine. The manuscript is well written and clearly organized.

Additional details related to the numbers and types of articles identified in the selective review performed would strengthen the paper.  How many articles were identified using the search criteria presented?  What process was used to narrow down from this original number?  How were the examples included in this manuscript ultimately selected? 

Moreover, were any articles identified that had contradictory findings from those presented? For example, were any studies identified that illustrate men in receipt of inadequate emergency care vs. women?  Are there certain instances when pain experiences are comparably greater among men vs. women? It was noted that studies suggest that men and women respond differently to pain, with women generally having a higher risk for clinical pain.  Perhaps there are examples here of men exhibiting greater pain in the face of certain types of medical issues that could be shared with readers? The literature reviewed and examples provided are helpful in understanding the myriad ways in which gender disparities are present in the healthcare context. However, additional details about literature with differing findings would aid readers in understanding the complete context of the issue.  For example, if no studies were identified that demonstrated contrary findings, this should be stated. If studies that demonstrated contrary findings had some major shortcomings, these could be briefly described. 

In the example of HPV vaccination, it would be helpful to also consider differences in vaccination between men and women given the vaccine is recommended for both and protects against cervical cancer as well as multiple cancer types (e.g., oropharyngeal, penile, anal) that affect men. Why do parents who decide against vaccinating their sons against HPV make the decision? Does their reasoning differ from parents who make this decision for their daughters or is it similar?

Minor issues to correct
- Provide full name of WHO and UN when first listing 
- repetition in final sentence of section 6.1 (although it is now known that a high vaccination rate against HPV in girls at the beginning of puberty more than halves the incidence of cervical carcinoma stated twice)

Reviewer 2 Report

Comments and Suggestions for Authors

An excellent choice of topic but I am not sure how much this review adds to existing literature. Is there anything new that came out of this review that we did not know before? The authors cite 5 systematic reviews in doing this study. What does this review add?

As this review is a narrative one consider describing it as such in title and throughout to emphasize that it is not a systematic review

How much of this study's conclusions are a result of this review? There are 5 references in the conclusion

Reviewer 3 Report

Comments and Suggestions for Authors

The review titled "The importance of gender-sensitive health care in the context of pain, emergency, and vaccination: a review" is interesting and well-written, yet it can be further improved.

  1. 1) The authors need to explain the choice of domains selected for inclusion in the review, namely "pain," "emergency care," and "vaccine care." The rationale behind choosing only two of them (pain and emergency care) is evident, but the role of vaccination is not clear.

  2. 2) It is not a common approach for a narrative review to include the Methods section. However, if the authors have opted to do so, they need to develop it further and provide more details on the filters used and the strategy of search in different databases. Inclusion and exclusion criteria also need to be mentioned.

  3. 3) The review lacks any tables or figures. I would advise the authors to add three figures presenting a schematic overview of each of the three domains covered by the review.

Round 2

Reviewer 2 Report

Comments and Suggestions for Authors

A very good resubmission. My only comment is that the conclusions should be the very last section and further developments should be before the penultimate section.

Reviewer 3 Report

Comments and Suggestions for Authors

WelL done!